

# Surface circulation properties in the Eastern Mediterranean emphasized using machine learning methods

Georges Baaklini[1,2], Roy El Hourany[3], Milad Fakhri[2], Julien Brajard[4], Leila Issa[5], Gina Fifani[1,2], and Laurent Mortier[1]

[1]Sorbonne University, UPMC Univ Paris 06 CNRS-IRD-MNHN, LOCEAN Laboratory, 4 place Jussieu,75005 Paris, France
[2]National Centre for Marine Sciences-CNRSL, P.O. Box 189, Jounieh, Lebanon
[3]Univ. Littoral Côte d'Opale, Cnrs, Univ. Lille, UMR 8187, Laboratoire d'Océanologie et de Géoscience (LOG), Wimereux, France
[4]Nansen Environmental and Remote Sensing Center, Bergen, Norway
[5]Department of Computer Science and Mathematics, Lebanese American University, Beirut, Lebanon

**Correspondence:** Georges Baaklini (georges.baaklini@locean.ipsl.fr)

**Abstract.**

The Eastern Mediterranean surface circulation is highly energetic, composed of structures interacting stochastically. However, some main features are still debated, and the behavior of some fine-scale dynamics and their role in shaping the general circulation is yet unknown. In the following paper, we use an unsupervised neural network clustering method to analyze the long-term variability of the different mesoscale structures. We decompose 26 years of altimetric data into clusters reflecting different circulation patterns of weak and strong flows with either strain or vortex-dominated velocities. The vortex-dominated cluster is more persistent in the western part of the basin, which is more active than the eastern part due to the strong flow along the coast, interacting with the extended bathymetry and engendering continuous instabilities. The cluster that reflects a weak flow dominated the middle of the basin, including the Mid-Mediterranean Jet (MMJ) pathway. However, the temporal analysis shows a frequent and intermittent occurrence of a strong flow in the middle of the basin, which could explain the previous contradictory assessment of MMJ existence using in-situ observations. Moreover, we prove that the Levantine Sea is becoming more and more energetic as the activity of the main mesoscale features is showing a positive trend.

## 1 Introduction

The Levantine Sea surface circulation is reigned by a complex mesoscale system composed of eddies, jets, and filaments interacting stochastically with each other (Özsoy et al., 1991). This basin lies on the easternmost part of the Mediterranean, bounded by the Cretan Archipelago, Asia Minor to the north, the Middle East to the east, and north-eastern Africa to the south. One of the main reasons behind its complex dynamic is its peculiar bathymetry, characterized by the presence of the Herodotus Abyssal Plain at 3000 $m$ depth. It is a deep and large sub-basin located in the southern part of the Levantine basin between Libya and Egypt. Yet, also other shallower sub-basins exist, such as the Lattakia, Cilicia, and Antalya. In the south of Cyprus, there is a notable presence of the Eratosthenes seamount, whose summit is about 700 $m$ deep. Furthermore, the mesoscale activity of the Levantine basin has a continuous and direct impact on physical and biogeochemical water properties,



where for example, currents transport chemicals originating from rivers and nutrient-rich waters into the oligotrophic open sea (Taupier-Letage et al., 2003; Lehahn et al., 2007; Escudier et al., 2016). Although the mesoscale activity is highly evolving in the Eastern Mediterranean, some of the mesoscale features are almost permanent and appear as separate building blocks

(Matteoda and Glenn, 1996; Rio et al., 2007). These well-defined blocks include the anticyclonic eddies of Mersa-Matruh, Shikmona, Cyprus eddies, and the cyclonic Rhodes gyre (see fig. 1) (Amitai et al., 2010; Larnicol et al., 2002; Gerin et al., 2009; Menna et al., 2012).

Several studies have aimed to characterize the surface dynamics of this basin, such as Hamad et al. (2005) or Taupier-Letage (2008), which provided a descriptive analysis of the Eastern Mediterranean sea surface currents using Sea Surface

Temperature images from satellites. Moreover, by using altimetric data, the long-term averaged Mean Dynamic Topography (MDT) (Amitai et al., 2010), current flow kinetic energy variability (Pujol and Larnicol, 2005; Menna et al., 2012), and eddies tracking (Mkhinini et al., 2014) allowed to analyze the long-term surface current and long-lived eddies. However, these studies showed average patterns, mainly emphasizing existing and permanent eddies activities, with less focus on interannual variability or other patterns, such as jets and weak flows.

Additionally, the debatable presence of the Mid-Mediterranean jet (MMJ, see fig. 1), a cross-basin flow of the AW, is still not answered, resulting in contradictory assessments. Some authors consider the MMJ as an artifact caused by the deviation of the coastal AW, driven from one eddy to another. In other terms, it is the result of the paddle-wheel effect due to the high mesoscale activity across the eastern Levantine. Indeed, no remarkable jets were observed during the Mediterranean Forecast System (MFS) program (Manzella et al., 2001) or during XBT campaigns (Horton et al., 1994; Zervakis et al., 2003) or by the sizeable

drifter data set released during the EGYPT/EGITTO program between September 2005 and July 2007 (Millot and Gerin, 2010; Gerin et al., 2009), or even when using high-resolution numerical models (Alhammoud et al., 2005). Nevertheless, other observational studies have shown a clear across-flow in the middle of the basin with an average speed ranging between 10-19 cm/s (Amitai et al., 2010; Poulain et al., 2012). More recently, SST anomaly satellite images, drifter tracks, and the geostrophic currents computed from the satellite Absolute Dynamic Topography (ADT) fields showed an occasional MMJ flowing toward

the north of the Lebanese coasts between 2016 and 2017 (Mauri et al., 2019). In Ciappa (2021), the investigation of SST and altimetry images between 2000 and 2015 considered that the MMJ is triggered by the surface cold water pushed by a northerly wind, reaching the northern periphery of the Libyo-Egyptian coasts and the high vorticity off the LEE that inject the MMJ in the middle of the eastern Levantine basin.

Actually, satellite altimetry is a widely-used observational tool for analyzing sea surface physical dynamics and the mesoscale

activity and provides a continuous coverage for more than 25 years (Ducet et al., 2000; Bosch et al., 2014; Fu and Cazenave, 2000; Hamlington et al., 2013; Willis, 2010). When such a large data set is available, Machine learning methods could be applied efficiently to characterize the surface circulation and identify patterns. One efficient method is the Self Organizing Map (SOM, (Kohonen, 2013)), a high-performance unsupervised clustering algorithm. It is used to classify and extract features, previously applied in different fields; In the oceanography, the SOM was applied to characterize the inter-annual, seasonal, and

event scale variability of wind and Sea Surface Temperature patterns (Richardson et al., 2003), and phytoplankton pigment variability (El Hourany et al., 2019). The tool was also efficient for the characterization of the coastal areas by combining





high-resolution numerical models with radars observations (Ren et al., 2020), or radar with ADCP dataset, such as in the West Florida Shelf (Liu et al., 2007), and the Long Island Sound tidal estuary (Mau et al., 2007). In the Mediterranean, SOM was applied in the Adriatic Sea using the HF radar measurements (Mihanović et al., 2011), and in the Sicily Channel, using 46 years of a high-resolution model (Jouini et al., 2016). This approach allowed to decompose the surface circulation in the Sicily Channel into modes reflecting the variability of the circulation in space and time at seasonal and inter-annual scales. SOM was also able to provide a prediction of the surface current in the shallow coastal area (Kalinić et al., 2017) and to identify phyto-plankton functional types in the Mediterranean Sea using a bioregionlization approach (El Hourany et al., 2019; Basterretxea et al., 2018).

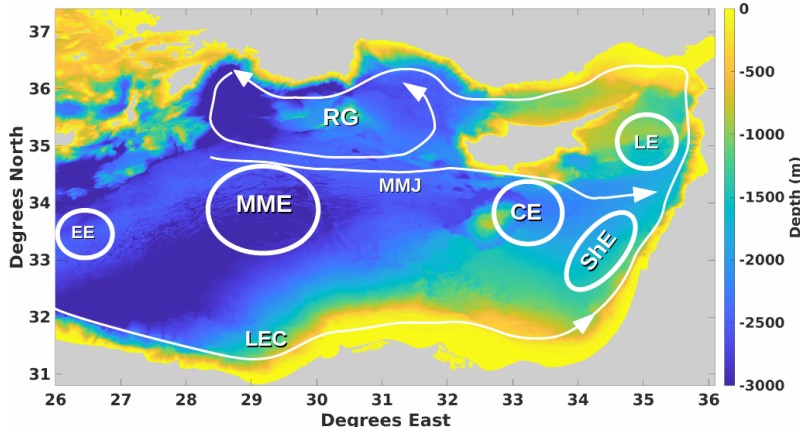

**Figure 1.** A schematic representation of the main structures present in the Levantine basin (RG: Rhodes gyre, LE: Lattakia eddy, EE: Egyptian eddies, LEC: Libyo-Egyptian current, MMJ: Mid-Mediterranean Jet, MME: Mersa-Matruh eddies (also known as Herodotus trough eddies), CE: Cyprus eddy, ShE: Shikmona eddies, LE: Lattakia eddy). All is overlayed on a bathymetry map.

In the light of the major gaps in characterizing the surface currents of the Levantine Sea and previous contradictory assess-ments, the present paper aims to improve the understanding of its surface dynamics and mesoscale structures using machine learning techniques. To reach this aim, we adapted the neural network clustering method, the Self Organized Map (SOM), and Hierarchical Ascendant Classification (HAC) as in (Jouini et al., 2016) that will allow the decomposition of the Levantine Sea surface velocity between 1993 till 2018 into different regimes; to estimate possible seasonal and inter-annual variations of the main mesoscale features. We also provide a possible explication of the reasons behind the contradictory assessment of MMJ's existence. We finally show the vorticity variation in the basin and the main factors affecting the apparition and persistence of the mesoscale features.

The paper is structured as follows: we start by presenting the data and the clustering method used to obtain the different clusters in section 2. Additionally, we present the approach used for the delimitations of the main mesoscale features zones. Results of decomposing the circulation are presented in section 3 and discussed in section 4. A conclusion is presented in section 5.





## 2   Data & Method

This section presents the data and methodology used for the restitution of the surface circulation regimes based on the concept
of Self-Organizing Maps (SOM) and the Hierarchical Ascendant Classification (HAC) methods. Then we present the approach
used to divide the basin into six geographical sub-regions (or boxes) that include the different mesoscale features.

### 2.1   Altimetry

The Altimeter satellite gridded Sea Level Anomalies (SLA) is estimated by Optimal Interpolation, merging the measurement
from the different altimeter missions available. This product is processed by the DUACS (Data Unification and Altimeter
Combination System) multimission altimeter data processing system. It processes data from all altimeter missions: Jason-3,
Sentinel-3A, HY-2A, Saral/AltiKa, Cryosat-2, Jason-2, Jason-1, T/P, ENVISAT, GFO, ERS1/2. To produce reprocessed maps
in delayed time, the system uses the along-track altimeter missions from products called $SEALEVEL\_*PHY\_L3\_MY\_008*$.
The SLA computation provides the Absolute Dynamic Topography (ADT) and the surface geostrophic currents.

The geostrophic surface velocity fields between 1993 and 2018, from the Herodotus abyssal plain and until the eastern-
most part of the Levantine sea, form the input layer of the SOM. In addition to the zonal and meridional components of the
geostrophic velocities, the fluid parameter of Okubo-Weiss ($OW$) is included in the input layer (see fig. 2). $OW$ measures
the relative importance of deformation and rotation at a given point. Positive $OW$ values indicate strain-dominated regions,
while negative $OW$ indicates vortex-dominated. Accordingly, $OW$ is a physical criterion widely used in the methods of eddies
detection.

$OW = s_n^2 + s_s^2 - w^2$, where $s_n$ and $s_s$ are the normal and the shear components of strain and the relative vorticity of the
flow defined respectively by

$$s_n = \frac{\partial U}{\partial x} - \frac{\partial V}{\partial y}; s_s = \frac{\partial V}{\partial x} + \frac{\partial U}{\partial y}; w = \frac{\partial V}{\partial x} - \frac{\partial U}{\partial y}; \tag{1}$$

### 2.2   The Self-Organizing Map (SOM)

The Self-Organizing Map (SOM) is an unsupervised neural network method used for data visualization. It projects higher
dimensional data into lower dimensional space leveraging topological similarity properties without requiring sample clusters.
By this method, multidimensional data are clustered into neurons automatically associated in orderly organisation, where
similar neurons are adjacent, and the less similar neurons are situated far from each other in the grid. This way allows obtaining
an insight into the topographic relationships of the initial dataset (Kohonen, 2013).

The SOM is structured in two layers: the input layer (in our case, a 3-D input layer composed of the zonal and meridional
components and the Okubo-Weiss parameter) and the resulting neuron grid. Each neuron, representing a cluster with data
presenting common characteristics, is associated with a referent vector obtained from a learning data set. Each vector of the
input layer will be attributed to the neuron with the closest Euclidean distance with the referent vector. This referent vector





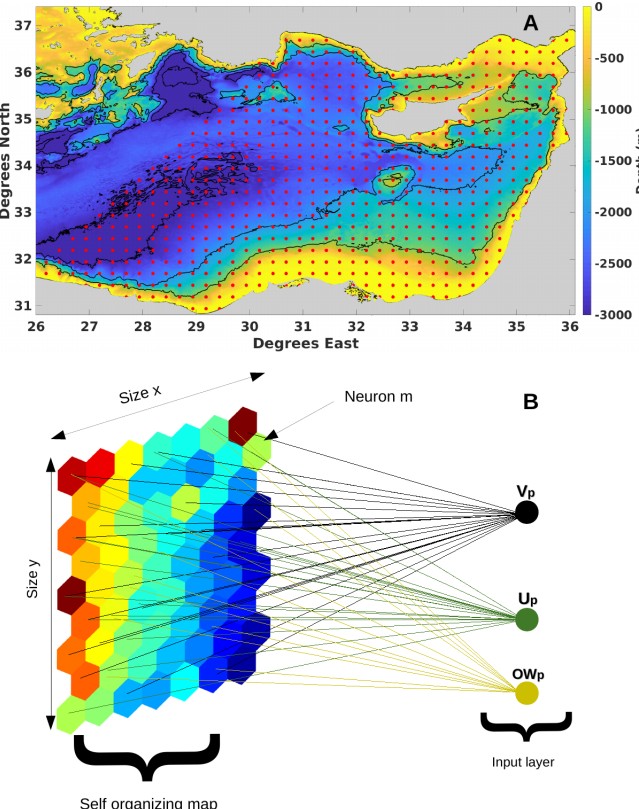

**Figure 2.** In the panel A, the red dots represent the coordinates of the grids providing the input data for the SOM. Each red dot (p) is composed of U and V components and the Okubo-Weiss parameter. The panel B is a schematic representation of the self-organizing map method: the input layer obtained from the input data and the adaptation layer composed of n neurons automatically associated in orderly fashioned order. Each neuron represents a set of U, V, and $OW$ that represents similarities.

is called the best matching unit (BMU), and its associated neuron is the "called" winning neuron. The determination of the referent vectors and the topological order of the SOM maps is done by minimizing the cost function:

$$\mathcal{J}_{SOM}^{T}(\mathcal{X}, W) = \sum_{zi \in D} \sum_{c \in SOM} K^{T}(\delta(c, \mathcal{X}(z_i))) ||z_i - w_c||^2 \tag{2}$$

where $c \in SOM$ represents the neuron index in the SOM, $\mathcal{X}(z_i)$ represents the allocation function that assigns each element $z_i$ of the input D to the corresponding referent vector $w_{\mathcal{X}(z_i)}$. $\delta(c, \mathcal{X}(z_i))$ represents the discrete distance on the SOM between a neuron c and the neuron allocated to observation $z_i$. $K^T$ is a kernel function parameterized by T that weights the discrete distance on the map and decreases during the minimization process. During the minimization of the cost function, the topological order is preserved, thus the more similar neurons are adjacent and the less similar neurons are situated far from each other.

To provide an equal weights distribution of the input parameters, the variables were normalized with their variances. Several




tests were conducted to determine the optimal size of the SOM map giving the best representation of the data. These tests were based on the capacity of each map to reproduce with the less error possible the initial dataspace of our input data. Based on that, we opted for a large SOM map of 1400 neurons.

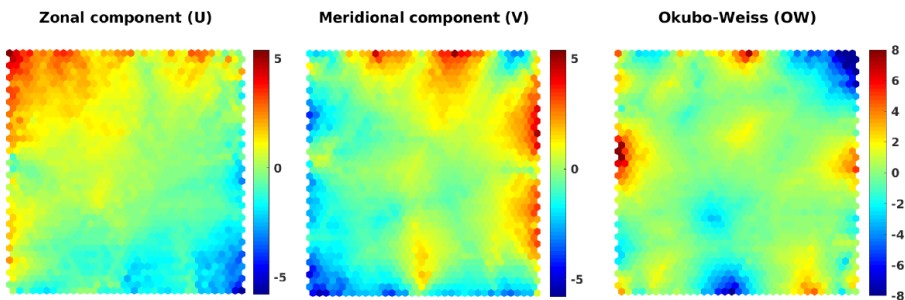

**Figure 3.** Topological maps showing the organization of the three variables (U, V, and $OW$) on the SOM after the training phase. Each map shows the recorded values by each neuron for the three variables.

After the training phase, the SOM is well organized, where there is a gradient of zonal and meridional velocities. This
distribution of U and V well-reflected the $OW$ variation. The $OW$ in SOM shows clusters of intense positive or negative values (see fig. 3). These extreme values represent the characteristics of a vortex (positive) or strain (negative) dominance.

## 2.3    HAC method

The SOM allowed classifying the velocity field into neurons that represent the different circulation patterns of the targeted grid, based on U, V, and $OW$. To simplify the representation of the physical processes obtained from the different situations
captured by each neuron , we applied the HAC (Hierarchical Ascendant Classification) to group these neurons into a reduced number of clusters. HAC is a cluster analysis that seeks to build a bottom-up hierarchy of clusters. From the initial partition containing the neuron groups of the SOM map, two neurons of the same neighborhood were clustered at each iteration. The used criterion was Ward's minimum variance method, which provides a partition that minimizes the within-cluster inertia (Randriamihamison et al., 2021) while respecting the topological organisation. As a result, the neurons were separated into five
different clusters (see fig. 4A). Consequently, clusters 4 and 5 (denoted as C4 and C5) were characterized by a negative $OW$, compared to positive $OW$ for C1 and C2, while C3 had an average $OW$ of 0 (C). MKE boxplots of the neurons in each cluster show that C3 had the weakest flow intensity, while C5 and C1 represented the highest MKE intensities, followed by C2 and C5 (B). In summary, C1 and C2 are clusters of a strong-flow with positive $OW$, C4 and C5 are clusters of a strong-flow with negative $OW$, while C3 is the cluster of the weakest velocities.

## 2.4    Studied area

The targeted area starts from the Herodotus abyssal plain until the easternmost part of the Mediterranean to analyze the activity of the most dominant features present in this area. Mersa-Matruh eddies (MME) (see fig. 1) are eddies generated by the coastal



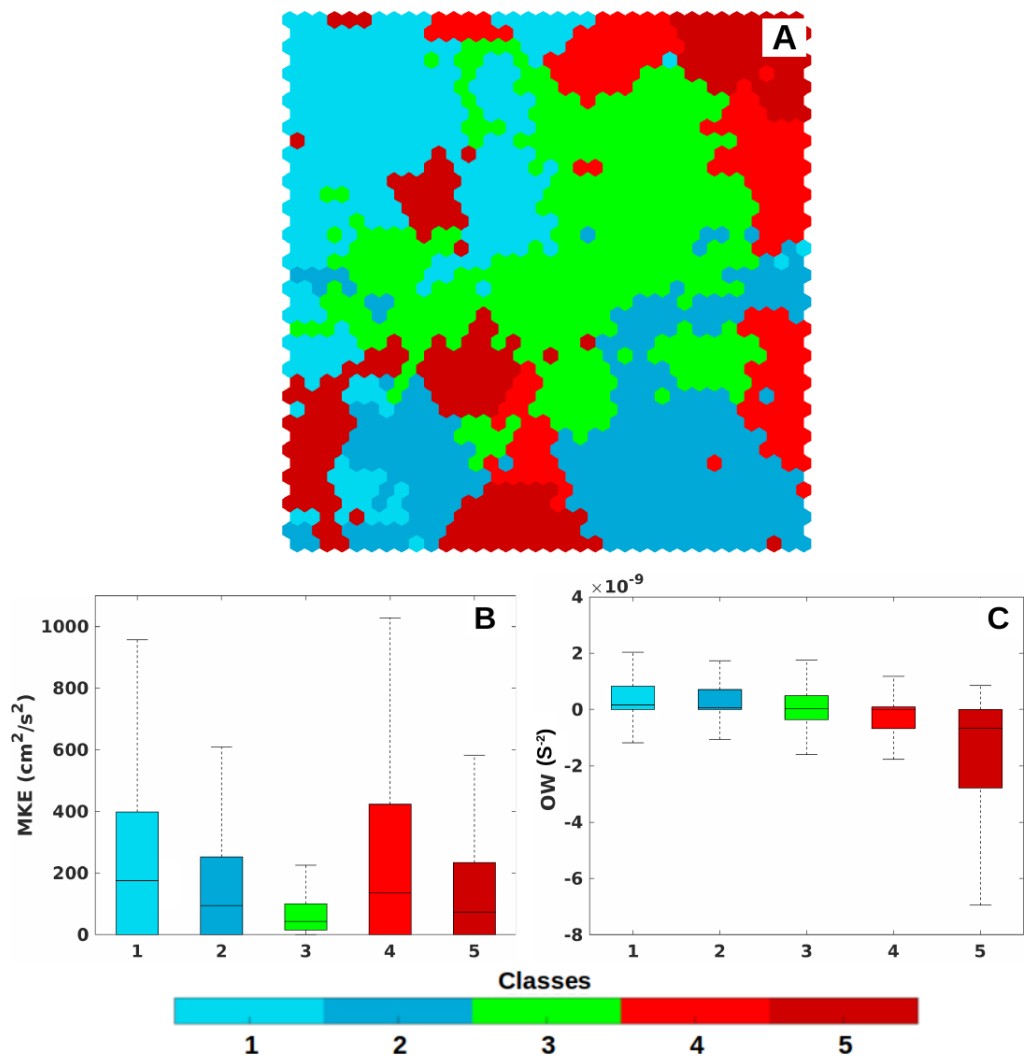

**Figure 4.** A the topological map of neurons representing the different clusters obtained from the SOM and HAC method. The resulting MKE and $OW$ values of each cluster are shown in the boxplots in panels B and C, respectively.

instabilities near the Eastern Libyan and Egyptian coasts. These eddies drift seawards while being guided by the isobath. Indeed, the 3000 $m$ Herodotus plain traps and prevents the coastal vertically-extended eddies from propagating further to the

east, thus causing an accumulation of these eddies in this area (Alhammoud et al., 2005; Elsharkawy et al., 2017). MME is one of the most dominant mesoscale features in the Eastern Mediterranean, where their Mean Kinetic Energy exceeds 300 $cm^2/s^2$ (Menna et al., 2012). In the easternmost part of the basin, Shikmona eddy (ShE) (see fig. 1) represents another complex system composed of several cyclonic and anticyclonic eddies with varying size, position, and intensity (Gertman et al., 2007; Menna et al., 2012; Mauri et al., 2019). Similar to MME, ShE is an area where previously formed eddies tend to accumulate and/or

merge (Hamad et al., 2005). Another important mesoscale feature existing in the eastern part is Cyprus eddy (CE) (see fig. 1).



It is an intense dynamic feature occurring in the open sea. Unlike the MME and ShE, eddies are formed in this area and do not accumulate (Zodiatis et al., 2005). We should note that other mesoscale structures exist in the eastern Levantine but are less frequently observed. Among these, we mention the "Lattakia Eddy" (LE) taking place between Cyprus and Syria. LE is a cyclonic eddy generated by the interaction of the northward current along the Lebanese and Syrian coasts with a Mid-

Mediterranean jet (Zodiatis et al., 2003), and/or between ShE and the coastline (Hamad et al., 2005), and/or by the topography (Gerin et al., 2009).

### 2.4.1 Definition of main mesoscale features regions

The eddies usually reveal elevations (anticyclones) or depressions of the sea surface. Accordingly, to study the variation in the activity of the main mesoscale eddies in the Levantine after decomposing the surface circulation, we delimited the spatial

area of each eddy by computing the Mean Dynamic Topography (MDT) map obtained from averaging Absolute Dynamic Topography (ADT) for over 26 years between 1993 and 2018. In fig. 5A , the iso-MDT closed lines show several eddies structures consistently present in the basin. The Positive MDT values, revealing anticyclonic circulation structures, were around the Eratosthenes seamount and the Herodotus Abyssal Plain depth. On the other hand, negative MDT values, hosting cyclonic circulation, were observed between Cyprus and the Asia minor and in the Shikmona area between the South of Lebanon

and Egypt. The representation of MDT variation (B) shows similar mesoscale structures with additionally two active zones, observed offshore the Lebanese coasts and the west of the Eratosthenes Seamount. Depending on these results revealed by the average and the variation of the Dynamic topography, we divided the basin's mesoscale activity into several boxes: the Beirut area off the Lebanese coast (Bei), Cyprus Eddy that includes the Eratosthenes seamount (CE), Mersa-Matruh eddy above the Herodotus plain (MME), Shikmona (Shik), and the Asia Minor Current area (AMC).

## 165 3 Results

In this section, we present the results of decomposing the surface circulation of the Levantine basin into a daily time-series of five clusters obtained by the HAC and SOM methods.

### 3.1 Temporal variation

The frequency variation of the five clusters in each of the selected boxes, Bei, Shik, MME, AMC, Nile, and CE, is seen in

fig. 6. This frequency variation reflects the percentage of pixels assigned to each of the five clusters in a designated box. As a result, except for C4 in the AMC, all the clusters permanently occurred, with different proportions highly variable with time in each box. Moreover, clusters frequency significantly varies from one box to another. Although clusters of strain-dominated flow (C1 and C2) were not frequent everywhere, C1 and C2 were frequently observed in MME and AMC, respectively, during all the period. Such a high frequency occurred at the expense of other clusters, especially the cluster of weak flow C3 that was

less observed in these two boxes. Regarding the vortex-dominated clusters (C4 and C5), C5 was the most frequent. C4 was quasi-absent in AMC and scarcely existed in the other boxes. When comparing between boxes, C5 was most frequent in the



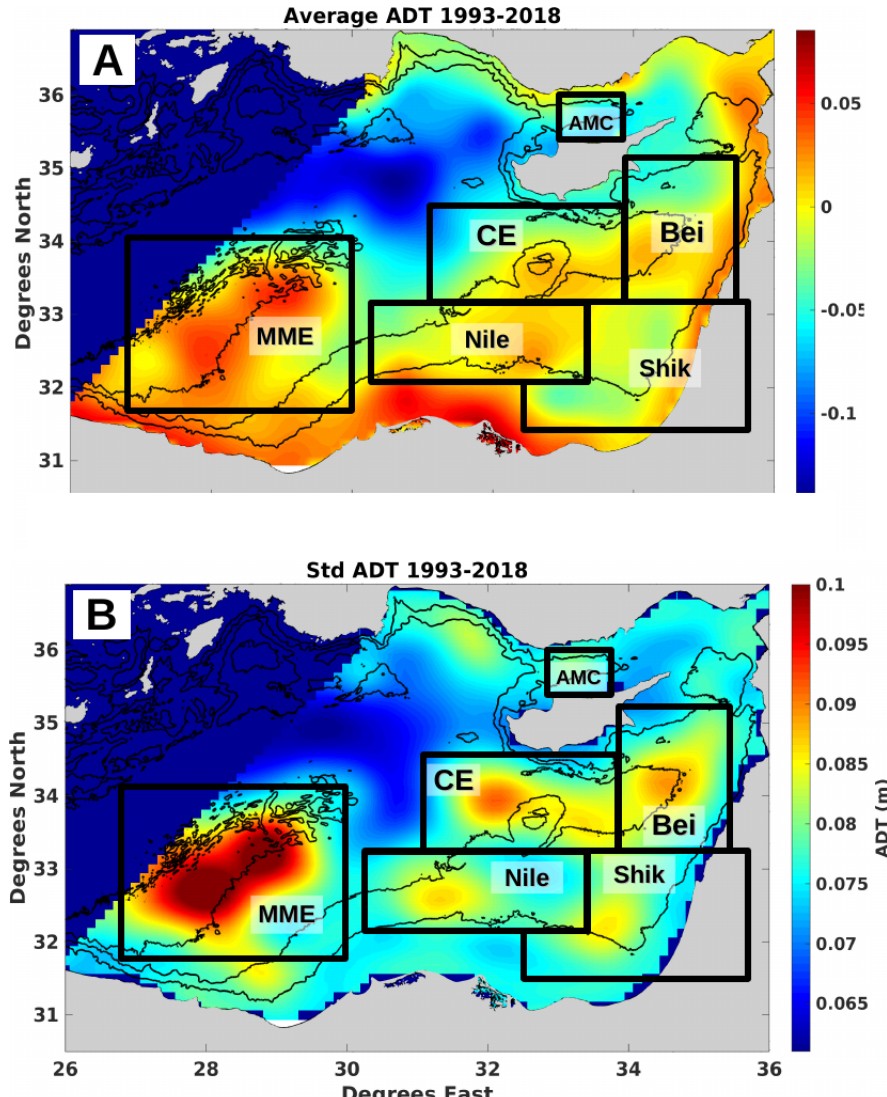

**Figure 5.** A shows Mean Dynamic Topography (MDT) between 1993 and 2018 in the Eastern Levantine, while B the standard deviation of the Absolute Dynamic topography (ADT) for the same periods. Both includes the names and borders of the six delimited sub-regions or boxes (MME: Mersa-Matruh Eddy, CE: Cyprus Eddy, Nile, Shik: Shikmona, Bei: Beirut, AMC: Asia Minor Current).

MME. Overall, all the clusters occurrences were highly fluctuating with time. To better analyze such a frequencies variation, we presented the daily dominant cluster in each box in fig. 7. On the one hand, C3 was the main cluster in CE, Nile, Bei, and Shik boxes. Moreover, between 1993 and 2000, C3 was almost exclusively dominating CE. On the other hand, C3 dominance was rare or quasi-nonexistent in the MME and AMC, where instead, C1 and C2 were, respectively, the most frequent clusters. While C2 dominance was not observed in MME, an increasing periodic C1 domination was observed in AMC, starting from 2000.





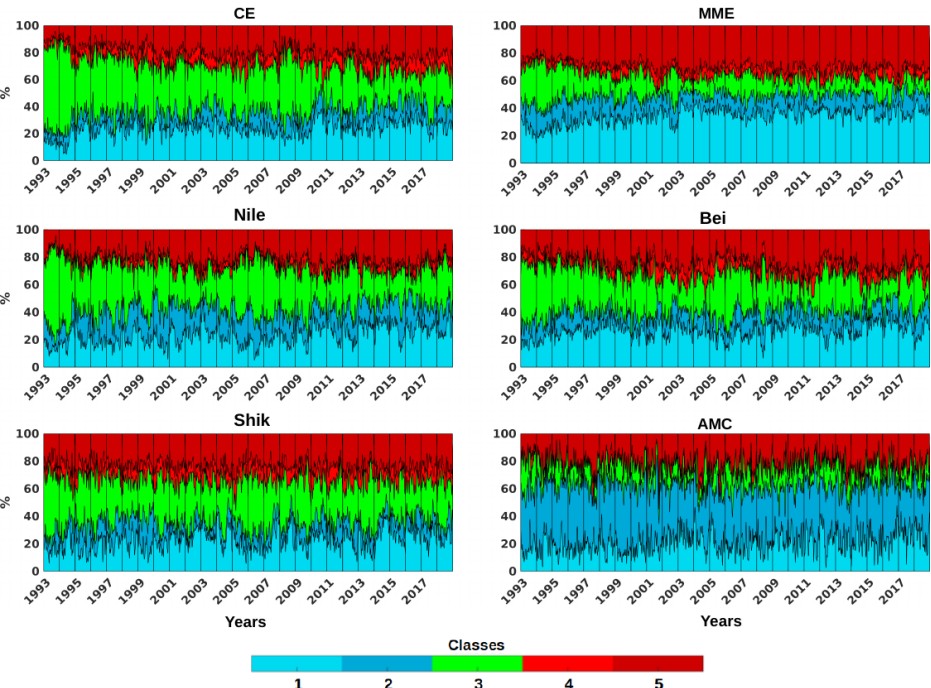

**Figure 6.** The daily variation of each cluster frequency in each of the selected boxes, between the beginning of 1993 and until late 2018. This frequency variation reflects the percentage of pixels that were affected to each of the five clusters in a designated box.

C5 was frequently dominating all the boxes, but intermittently. This dominance was rare in the CE, Nile, and AMC, compared to the MME, Bei, and Shik. Indeed, in these last three boxes, C5 dominance was frequent and showed a long lifetime that could last for several months, such as observed in Bei in 2003 and 2014 and Shik in 2005.

These results showed that MME and AMC are two zones of a special regime of flow. This latter is represented by clusters of intense current, the so-called C1, and C2. The other boxes are zones of relatively weaker currents. In all the boxes, there were sporadic events of intense eddy activity, exhibited by the intermittent periods of C5 dominance.

To quantify the flow intensity variation, the daily mean kinetic energy of the mean flow per unit of mass (MKE) is computed in each box (see fig. 8). It shows that the lowest MKE was observed in the boxes where C3 dominated the most (CE, Nile,

Shik, and Bei). In these boxes, MKE values were less than 150 $cm^2/s^2$. On the other hand, the highest values were in AMC and MME, with values consistently exceeding 300 $cm^2/s^2$. Hence, the boxes dominated by C1 and C2 had MKE values two times higher than the values observed in boxes dominated by C3.

The tendency of the dominant cluster could change with time, where for example, previous results show that between 1993 and 1997, C5 was rare in Bei before being more frequently observed as a dominant cluster. Figure 9 shows the seasonal variation

of the C1, C3, and C5 averages frequencies in all the boxes between 1993 and 2018. The general trend of C5 frequency was increasing with time. The most intense C5 positive tendency was in MME, where C5 increased by 10 % in 26 years. In all the seasons, the C5 frequency average in MME increased from 25 % in 1993 to 35 % in 2018. There was a similar increase in all





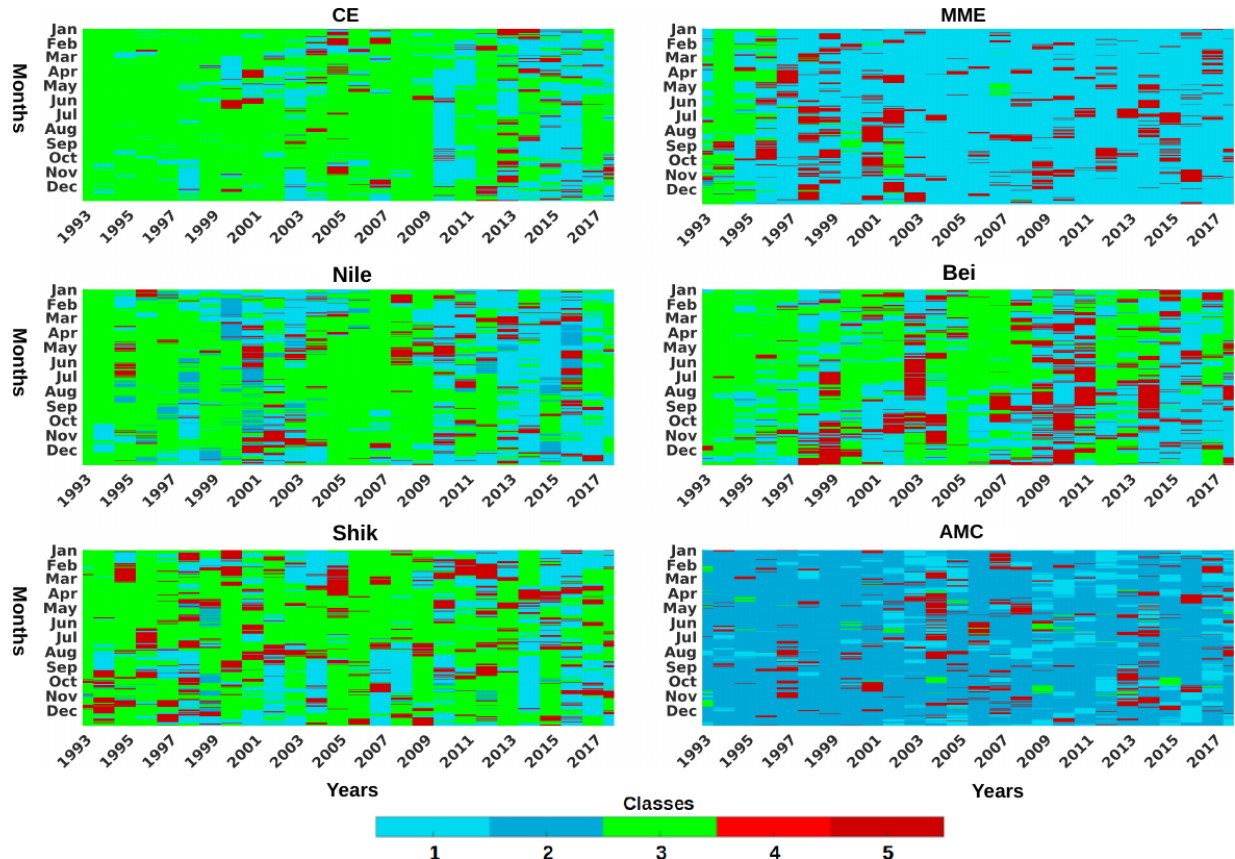

**Figure 7.** The daily dominant cluster in each box from 1993 and 2018.

the seasons in the Bei, MME, and Nile boxes. The weakest increasing tendency was in AMC and Shik. In terms of seasonality, there was no significant difference of C5 frequency between seasons, except in CE, where in summer and fall, the values were

higher with values closer to the MME. MME registered the highest seasonal average frequency of C1, meaning that this box is the most dominant in terms of intensity and vorticity (C5 and C1). In terms of trends, C1 showed similar results to C5. MME and Nile boxes showed the sharpest increases, followed by AMC, Nile, CE, and Bei, while C3 was continuously decreasing in all the boxes.

     These results show that the activity of the dominant mesoscale is increasing with time. Previous altimetric data observations

from 1993-2003 revealed increasing variability of the Mediterranean Sea activity that is maximal in the Levantine Sea, especially in the Mersa-Matruh area, where increasing energetic structures were observed (Pujol and Larnicol, 2005). Studies have discovered evidence that eddies are becoming more energetic. This increase was related to several reasons, such as the changes in winds or large-scale horizontal temperature gradients or the changes in the shear of the ocean currents (Martínez-Moreno et al., 2021). This current evolution is well highlighted where C3 is being progressively replaced by C1. Nevertheless, we

should not exclude the impact of the altimetry increasing accuracy in describing the surface circulation, with an increasing



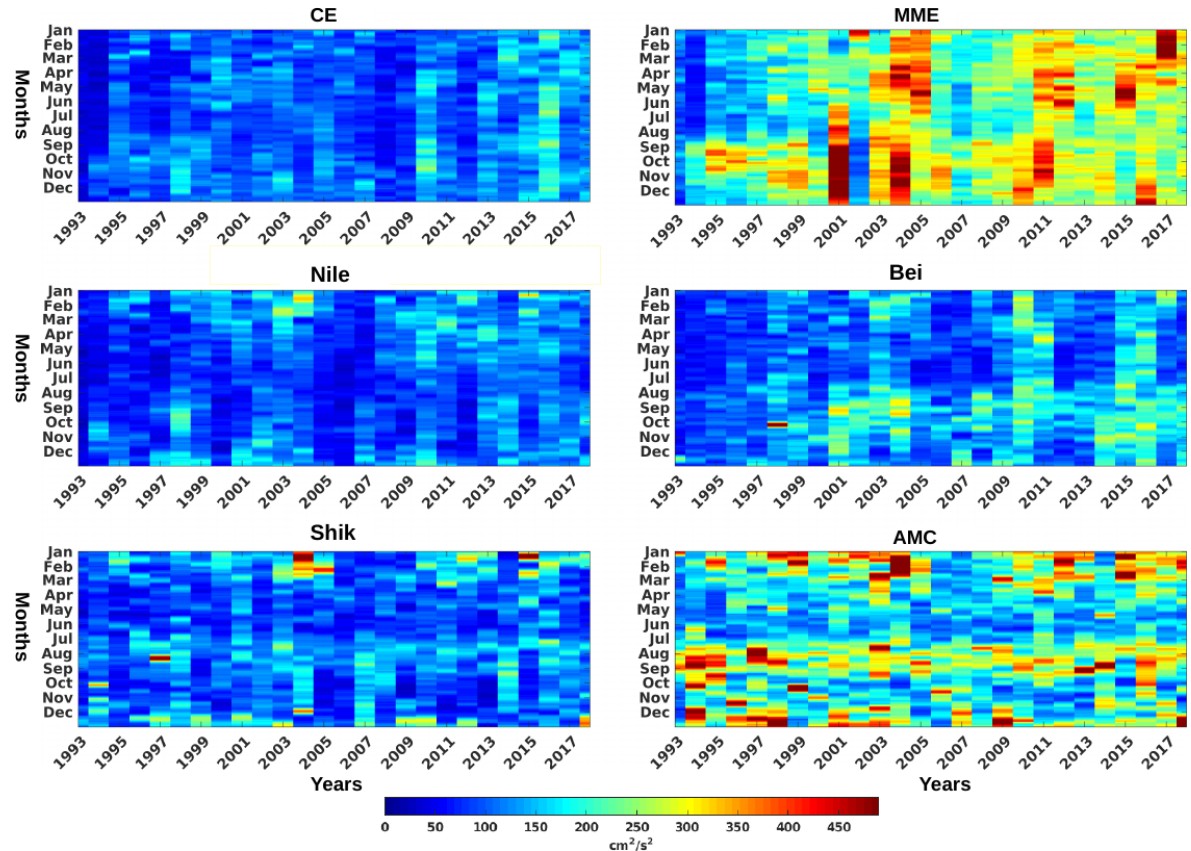

**Figure 8.** The variation of the daily average Kinetic energy (MKE $cm^2/s^2$), in each box from 1993 to late 2018.

number of satellite tracks in both time and space. Indeed, combining satellite altimeters proved its efficiency in providing a better representation of the mesoscale activity (Pascual et al., 2006; Amores et al., 2019). In terms of seasonality, there was no significant difference of C5 frequency between seasons, except in CE, where in summer and fall, the values were higher with values closer to the MME. Although previous studies have mentioned that eddies in MME are mainly formed in sum-
mer and spring (Hamad et al., 2005; Mkhinini et al., 2014), there was no clear seasonality of C5 frequency. It is explained by the frequent eddies apparition and their relatively long lifetime. Consequently, eddies permanently occur in the MME box (Taupier-Letage, 2008).

It should be mentioned that C2 and C4 did not show a clear tendency. However, C2 was only significantly present in AMC with values around 40%, while in all other boxes, the frequency was less than 20% (see fig. 10).

**3.2 Spatial analysis**

The spatial variation of clusters' frequencies is shown in fig. 11. The along-slope coastal circulation in the eastern Levantine did not reveal consistent clusters frequencies. The high kinetic energy clusters, C1 and C2, were highly persistent off the

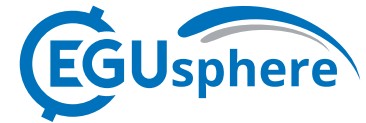

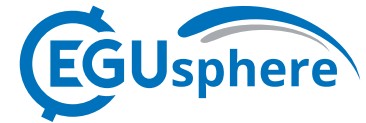

**Figure 9.** The seasonal variation of the C1 (high kinetic energy), C5 (high vorticity), and C3 (low KE and low vorticity) average in each box and their resulting linear regression.

Libyo-Egyptian coasts and between Turkey and Cyprus, respectively, with a frequency exceeding 50%, before being replaced by the weak flow cluster C3 in the easternmost part of the basin. The C5 predominance is mostly situated in the Western part, revealing that the mesoscale activity is more intense in that area. No clear jet was observed in the middle of the basin, including the MMJ pathway, where C1 dominance was disconnected by high C3 occupancy, more specifically between 30 and 32°E (see C1 in fig. 11).

### 3.3 Mid-Mediterranean jet (MMJ)

To further investigate the time evolution of the potentially existing MMJ, we present in fig. 12A a Hovmoeller diagram (along longitude 31.5625 °E) that shows the temporal variation of the clusters along its potential path. The longitude selection was based on the previous results showing area dominated by weak flow cluster C3 in the MMJ potential pathway between the Eratosthenes Seamount and Cyprus coast (see C1 frequency in fig. 11). Although frequent C3 domination, there were years





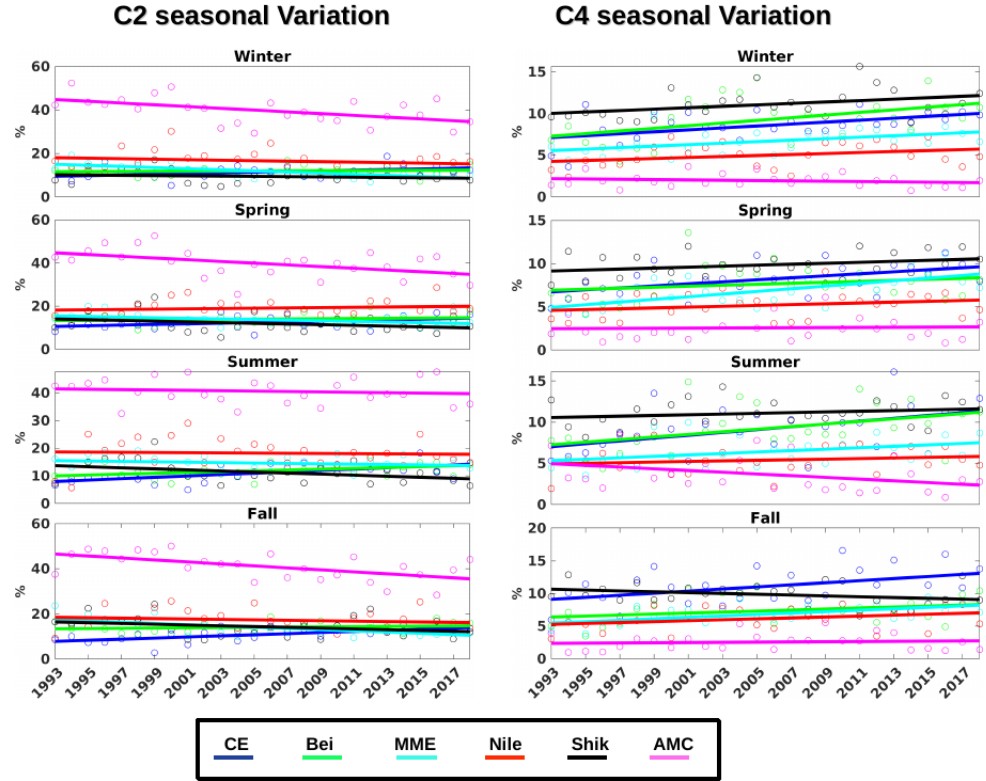

**Figure 10.** The seasonal variation of the C2, C4, and C3 average in each box and their resulting linear regression.

where clusters revealing higher KE were frequently occurring, such as C1 in 2015 or C5 between 2013 and 2014. An example of this variation is presented in fig. 12B, which shows the clusters frequencies distribution during the EGYPT/EGITTO

(September 2005–July 2007) CINEL campaigns (September 2016-August 2017) that provided in-situ drifters observations in MMJ potential pathway. During EGYPT/EGITTO campaign, C3 frequency was more than 50 %, and the other clusters did not exceed 15 %, while through the CINEL campaign, clusters of stronger KE increased at the expanse of C3, decreasing to 27 %. All these results reveal that in the middle of the basin, the MMJ is present, but sporadically and not as one of the most remarkable features, where the strong flow clusters were frequently observed but masked by C3, the dominant cluster there. In

Millot and Gerin (2010), the drifter data set used from the EGYPT/EGITTO campaign that showed no distinct jet was between 2005 and 2007, a period of C3 domination. On the other hand, in Mauri et al. (2019) the drifter tracks that showed a clear MMJ crossing the basin from west to east was between late 2016 and 2017, a period when there was a sharp shift in C3 frequency, decreasing by more than half compared to EGYPT/EGITTO. The Hovmoeller diagram also showed that MMJ variability was without any seasonal or periodic nature, in agreement with Ciappa (2021).





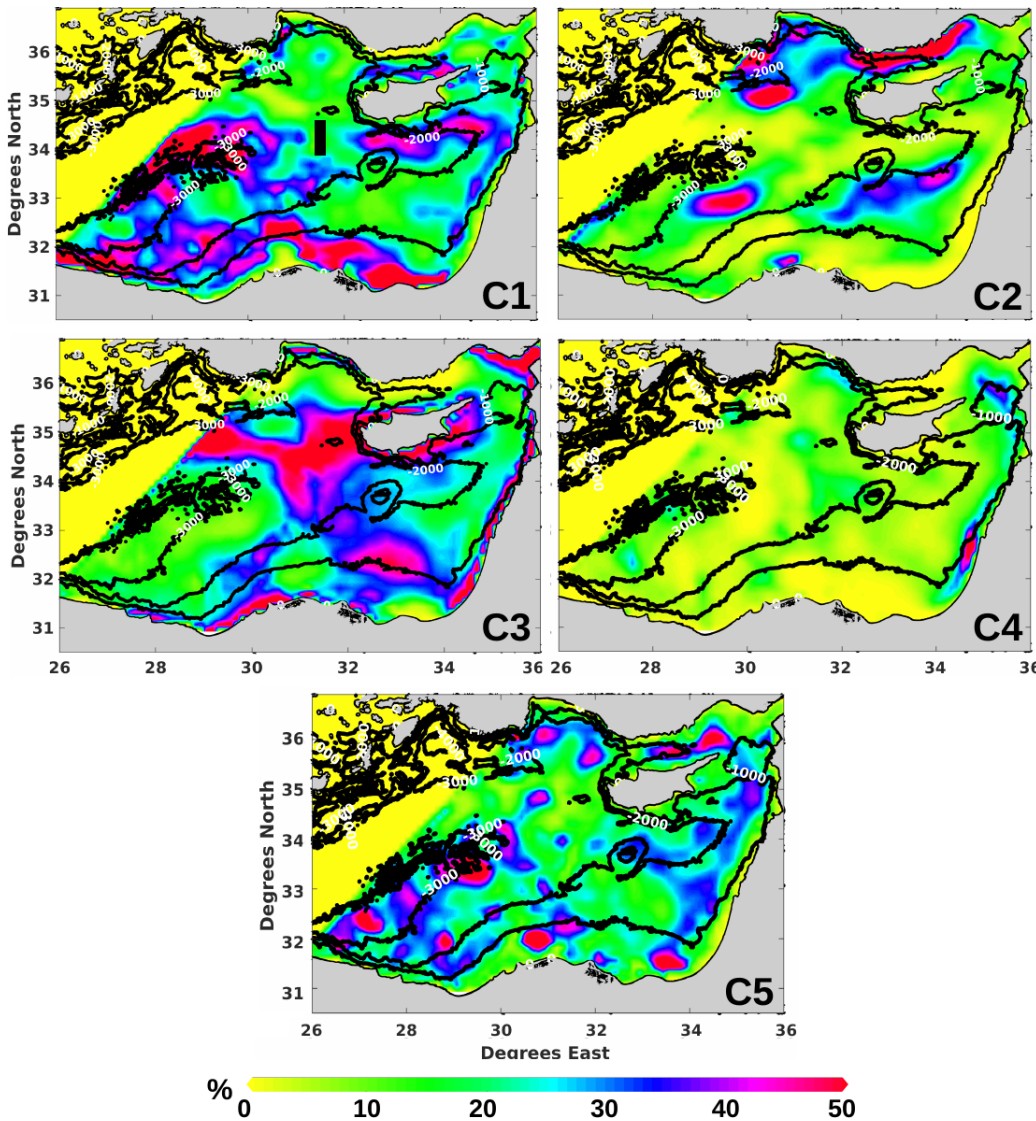

**Figure 11.** Percentage of C1, C2, C3, C4, C5 occurrence between 1993 and 2018, overlayed on the main bathymetric iso-lines of -1000, -2000, and -3000 $m$. The dark line in the C1 subplot shows the position of the Hovmoller diagram in sec. 3.3.

## 3.4 Vortex-dominated cluster analysis

Both of C4 and C5 are vortex-dominated clusters. However, the previous results showed that C4 is a peripheral cluster scarcely observed, dominating only very few pixels close to the coast. C5 was the main cluster that mainly reflected the eddies' presence. Here we present a more detailed analysis of the C5 evolution that reveals eddies activity in the Levantine sea.

Figure 13A shows the spatial distribution of C5 whose occurrence exceeded 40 % (C5P). The highest persistent C5P number was at the borders of the Herodotus plain. In addition, another group of C5P occurred in areas of the extended continental





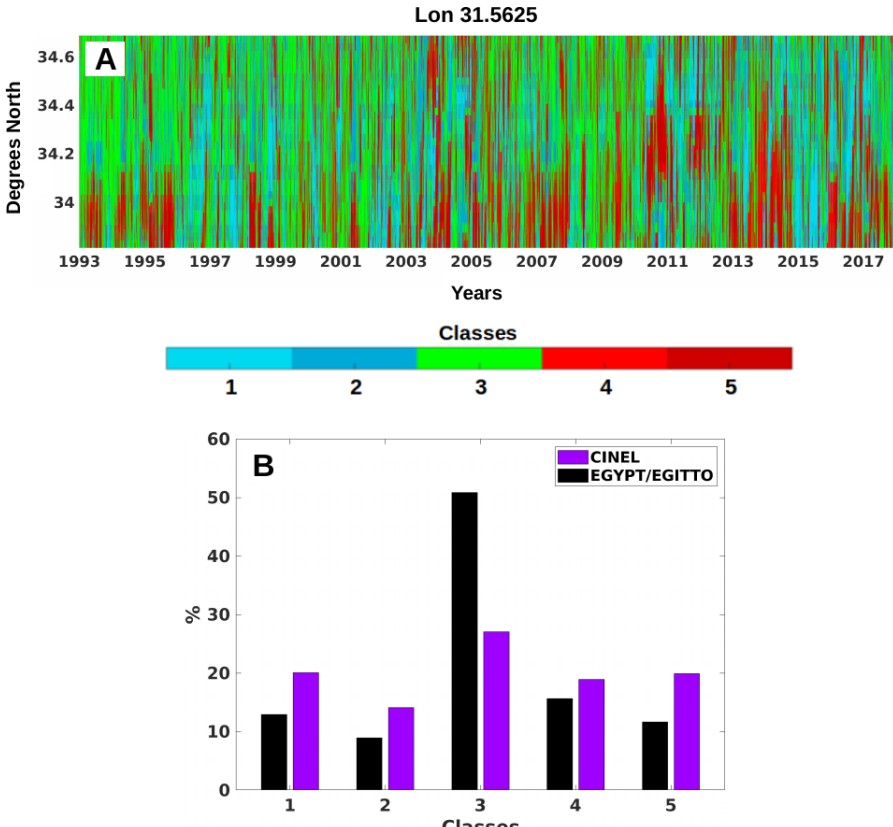

**Figure 12.** Panel A shows the Hovmoeller diagram of daily clusters variation along the MMJ potential pathway, at the longitude 31.5 °E. The resulting clusters frequencies during the EGYPT/EGITTO (September 2005–July 2007) and CINEL campaigns (September 2016-August 2017) are shown in panel B.

shelf, more precisely the shelf located offshore Egypt and between Cyprus and Turkey in the northern part of the basin. A small number of C5P followed the bathymetric iso-bath of 1000 $m$ in the western part of the basin. On the other hand, C5P was absent in the eastern part of the Levantine. In the panel B, we present the variation of C5P numbers regarding their distances with the main bathymetric structures (iso-lines of 1000, 2000, and 3000 $m$). More than 80 % of C5P were located at a distance

less than 60 $km$ from these main features. In the zones of extended continental shelf, such as offshore the Libyo-Egyptian coasts, there is a strong flow dominated by C2 and C1. Previous studies have shown that a current becomes unstable when wider than the bathymetry. Thus it favors the eddies formation conditions (Wolfe and Cenedese, 2006). The C5P absence off the Lebanese coasts is explained by the tight continental shelf almost absent. Indeed, the weak current dominated by C3 near the coastline is not strong enough to permanently create instabilities. The group of C5P observed in MME is due to the Herodotus

abyssal plain impact. The vertically extended eddies pinch off from the coast and propagate to the east before being trapped, thus accumulating eddies (Alhammoud et al., 2005; Elsharkawy et al., 2017). These results show that the main bathymetric features could potentially influence eddies creation and persistence in the Levantine basin.




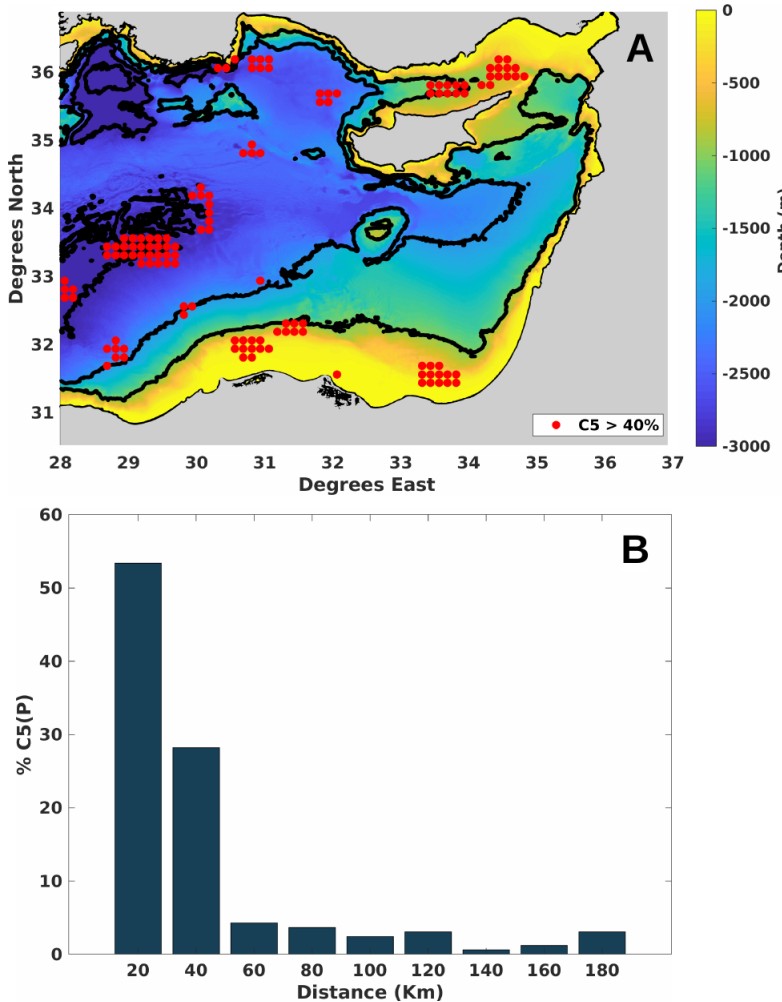

**Figure 13.** In panel A, the red dots show the positions of C5P, which are the pixels where C5 existed more than 40 % of the time. The dark line represents the main bathymetric iso-lines (1000, 2000, and 3000 $m$). Panel B shows the variations of C5P frequency compared to their distance with these iso-lines.

# 4 The effect of assimilation on the clustering

Because of the coarse resolution in both space and time of the altimeters, and to the fact that mesoscale structures move
continuously, eddies can be missed or artificially created, smoothed, misplaced, or be aliased into larger features compared to the true eddies (Mkhinini et al., 2014; Ioannou et al., 2017; Amores et al., 2019). To evaluate if the previously observed results are sensitive to the accuracy and the resolution of the altimetric data, we used a method to assimilate drifters with altimetry to obtain an improved representation of the surface circulation. We took advantage of three drifters, released during the CINEL campaign in 2017, and trapped in the Cyprus eddy for several months between $7^{th}$ of March (denoted as $D_0$) and until 31 $^{st}$ of





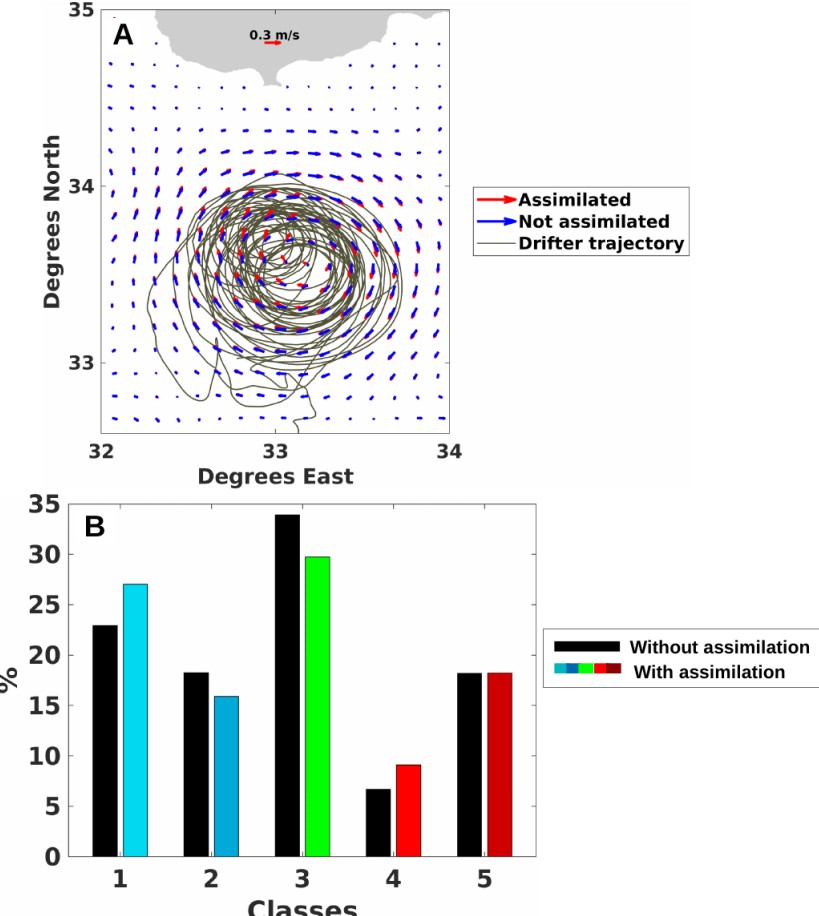

**Figure 14.** Panel A shows the average velocity field obtained before (blue) and after assimilating (red) the drifters' trajectories represented by the grey lines, circulating in CE from the start of March until late July. Panel B represents the percentage of pixels assigned to the clusters from 1 to 5 before (dark bars) and after assimilation (colored bars).

July 2017 (denoted as $D_{end}$) to assimilate them with altimetry (drifters OGS id: 5321, 5318, 5312). It is a variational approach, for which the altimetry is corrected by matching observed drifters' positions with those predicted by an advection model (Issa et al., 2016). The method proved its efficiency in providing an improved representation of the surface circulation along and around the drifters' trajectories, especially in high vorticity area (Baaklini et al., 2021).

The average velocity field shows differences in CE after assimilation (see fig. 14A). To evaluate assimilation impact on the
surface circulation decomposition, we assigned each velocity observation, before and after being affected by the assimilation, to a cluster using the trained SOM and HAC methods. In the panel B, we compared between the resulting clusters frequencies of the observed velocities before and after assimilation. It reveals that the assimilation slightly modifies the clusters proportions, except for C5, where C1 and C4 increased while C2 and C3 decreased. Hence clusters that are characterized by a strong MKE flow slightly increased at the expense of the other. Indeed, although altimetry is a coherent tool, assimilation showed



that eddies intensities in the Levantine could be underestimated. Besides that, the surface circulation could be less accurate closer to the coast, where previous studies have shown a declining altimetry accuracy in the Levantine basin (Fifani et al., 2021). Accordingly, the use of higher Spatio-temporal resolution products, such as accurate models, or the upcoming altimetry missions like SWOT that will provide more reliability close to the coast (Morrow et al., 2019; d'Ovidio et al., 2019; Barceló-Llull et al., 2021), will improve the decomposition of the surface circulation by our method, without changing the main

conclusions.

## 5   Conclusion

In this study, we analyzed the surface circulation of the Levantine basin using the SOM + HAC method that allows decomposing 26 years data set of surface geostrophic velocities into five clusters representing the different surface current flowing types. By tracking the clusters variability, we showed that the surface circulation is complex and divided into several energetic

boxes. We highlighted the increasing mesoscale activity in the basin, where these eddy-rich boxes are showing a positive trend with time. The cluster of weak flow is being progressively substituted by those of higher kinetic energy and vorticity, and thus the Levantine Sea is becoming more and more energetic. We were able to show the sporadic occurrence of the MMJ, which could explain the contradictory statements about the MMJ's existence. We highlighted the crucial role of bathymetry and the coastal flow intensity in increasing instabilities and eddies formation in the Levantine sea. Accordingly, the most

persistent eddies occurred in areas characterized by a strong coastal flow and an extended continental shelf or around Herodotus Abyssal plain, explaining thus the disproportions of eddies frequencies and persistence in the Levantine. It is a promising method that will undoubtedly benefit from more accurate and higher resolution of future altimetric missions. Also, it could be associated with other parameters such as Sea Surface Temperature and chlorophyll to study the interactions between the physical and biogeochemical water properties. Further work should expand the studied area to the entire Mediterranean to

investigate whether these increasing trends are only observed in the eastern Levantine or extend to a larger scale.

*Code and data availability.*   The altimeter products were produced by Ssalto/Duacs and distributed by AVISO, with support from CNES (http://www.aviso.altimetry.fr/duacs/).

Bathymetric data used in the figures are GEBCO data with 400 $m$ resolution, available at https://download.gebco.net/.

Drifters data were provided from: doi:data/10.6092/7a8499bc-c5ee-472c-b8b5-03523d1e73e9.

SOM algorithm was applied on Matlab, using the software library SOM Toolbox 2.0 Copyright (C) 1999 by Esa Alhoniemi, Johan Himberg, Jukka Parviainen, and Juha Vesanto and accessible at https://github.com/ilarinieminen/SOMToolbox.

*Author contributions.*   Conceptualization, GB, RE, and LM. Methodology, GB, RE, LM. Software, GB, RE, and GF. Validation, GB, RE, MF, JB, LI, GF and LM. Formal analysis, GB, RE and LM. Investigation, GB, RE, LI, JB and LM. Resources, MF and LM. Data curation, GB, RE, and GF. Writing—original draft preparation, GB, Writing—review and editing, RE, JB, LI, LM, and MF. Visualization, GB, RE



and GF. Supervision, RE, JB and LI . Project administration, MF, and LM. Funding acquisition, MF, and LM. All authors contributed to manuscript revision, read, and approved the submitted version.

*Competing interests.* The contact author has declared that neither they nor their co-authors have any competing interests.

*Acknowledgements.* We would like to acknowledge the National Council for Scientific Research of Lebanon (CNRS-L) for granting a doctoral fellowship to Georges Baaklini. This work was partially funded by ALTILEV (in the framework of the PHC-CEDRE project), and
O'LIFE programs (Observatoire Libano-Français de l'Environnement). We thank Milena Menna for her contribution in providing the drifters data.



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
