# Peer review of "Surface circulation properties in the Eastern Mediterranean emphasized using machine learning methods"

_EGUsphere, 2022_

## Author Comment (AC1)

Dear Reviewer,

We appreciate your precious time in reviewing our work. Your valuable and insightful feedback has improved the manuscript. We have carefully considered your comments below in black and tried to address them point-by-point in red. We refer to a modified pdf where all the modifications are highlighted in red.

1. My main comment is that the sections of the paper could be re-aranged in a more traditonal way while keeping the same content.

For example, Section 3 - Results, is actually a "Results and Discussion" section since the results are discussed with it ( and at the moment, there is not a real Discussion section)

> We agree with the reviewer, and therefore we replaced the section "Results" with "Results and Discussion" as we discuss the results in the same section.

2. Section 4 is a bit floating... I wonder if it is really needed (it is interesting, but personally I would remove since it is only based on a single deployment of 3 drifters).

> In this analysis we are not testing the assimilation accuracy method that has been developed for this region, but we are evaluating the robustness of our classification in case of eventual data correction. This analysis showed that the clusters are consistent regardless the used data (corrected or not). After correction, the classification shows no significant change in the clusters.

3. There are also a lot of figures. They are all discussed and potentially interesting, but it is a lot…

> We reduced the figures numbers by:
>
> • Merging figures 1 & 2
>
> • Merging 3 &4
>
> • Put figures 8 & 10 in appendix

-- Other Specific comments --

Introduction and Figure 1: An improved larger map with the political boundaries is needed. This will help the reader not familiar with the region.

> We modified the old figure 1 by showing a larger map with the political boundaries. We also merged figures 1 and 2 to reduce the number of figures.

Paragraph starting L.35: This seems a key aspect and could be introduced better.

> We added a sentence (line 35) (Additionally, there is a previous contradictory assessment of the presence of the Mid-Mediterranean jet (MMJ, see fig. 1A) (Ciappa, 2021), which is described as a surface current meandering across the Levantine Sea)

- L.35: "debatable" (is there any reference to this?)

> We added Ciappa 2021 ( line 35)

- L. 50: 25-yr of altimetry: maybe cite "International Altimetry Team (2021), Altimetry for the

future: Building on 25 years of progress. Advances in Space Research. doi: 10.1016/j.asr.2021.01.022."

> Done (line 51)

- The acronym SOM is spelled out several times (L.53, 79, 97, 98, …)

> We removed all the spelled acronym

- L.53: The sentence starting by "It is use to" is badly constructed.

> We replaced the sentence by: One efficient method is the Self Organizing Map (SOM, (Kohonen, 2013)) … (see line 52)

- L.69: replace "till" by "and", and remove semi-colon ";"

> Done

- L.86: Mention how the altimetry product was accessed (e.g. Copernicus? accessed date? doi? Citation?) :

> Done (we added link as well as the spatial resolution in line 87) . "The velocity field spatial resolution is 0.125∘ × 0.125∘ (available on http://www.aviso.altimetry.fr/duacs/)."

- Figure 3: Should we see a spatial map here? Are the pixels organized as in Fig 2?

> This is a representation of the SOM, each subunit of this map represents neurons, and the organization of the neurons represents gradients that can be seen in the data. We added a spatial map to show how the clusters are projected on a daily geographical map (see figure 2G)

- L.134: Would be nice to have a better description here of the clusters. Later you say "high EKE" (clusters 1-2) or "high vorticity" (clusters 4-5) and this is useful. I would mention it here too.

> We added a sentence that summarizes all the cluster in line (133- 134). "In summary, C1 and C2 are clusters of strong-flow with high vorticity (high MKE and positive OW ), C4 and C5 are clusters of strain dominated strong-flow (negative OW and high MKE), while C3 is the cluster of the weakest velocities."

- Section 2.4: I wonder if the study area should appear of the beginning of the method.

> Thank you for your suggestions, however, we preferred to leave the organization as is, because the definition of these regions is independent of the development of the method and is used to analyze mesoscale features.

- L.136: wrong construction with the sentence.

> Corrected

- Figure 4, 6, etc.: I would choose a different color scheme for the clusters. The 2 blues and 2 reds

are too similar (I printed the manuscript and you can't tell the difference). The colormap from Figure 11 would be best, for example.…

Thank you for your suggestions. We agree that the 2 reds and 2 blues were similar. To better distinguish between them, we changed the colors of clusters 1 and 5.

- L.138: "guided by the isobath" -> say which one.

We added the 3000 m isobath (see line 138)

L.160 and L.162: What you call "variation" is actually "standard deviation" and should be call this way.

Done we replaced by standard deviation (line 160 and 162)

- L.161: Replace "Depending" by "Based"

Done

- Section 3: It is really Results and Discussion...

We replaced Results by Results and Discussion (line 165)

- L.204 (whole paragraph): Maybe discuss a little bit more how the changes in satellite and sensors may have impacted (e.g. any abrupt changes corresponding to satellite changes?). Maybe also a good place to put your current Section 4 if you decide to keep it…

Done (We added  Satellite along-track sampling accurately estimates the SSH … in line 211)

- L.221: Not sure how to interpret the sentence starting with "The along-slope…"

Because the previous sentence was explicit, we replaced it with "The intensity of the along-slope coastal flow showed (line 225) ".

- L.222: "C1 and C2" -> If I interpret correctly, it is mostly C2… (C1 in Egypt C2 Cyprus) To

We replaced the old paragraph with another one between lines 223 and 225 that mentions the areas of C1, C2, and C3 persistence.

- L.223: Here a map with political boundaries would help too.

We agree with the reviewer, therefore we added the political boundaries in Figure 1.

- L.230: "along its potential path" -> along or accross?

Across (line 231)

- L.254: You mean their distance to the "closest" isobath between 1000, 2000, or 3000m? (same comment for Fig 13 caption).

- L.280: Reasonable to conclude that is could be underestimated while only comparing a single event with 3 drifters? See before

See answer to main comment number 2

---

## Author Comment (AC2)

Dear Reviewer,

We appreciate your precious time in reviewing our work. Your valuable and insightful feedback has improved the manuscript. We have carefully considered your comments below in black and tried to address them point-by-point in red. We refer to a modified pdf where all the modifications are highlighted in red.

Main concerns.

The work is based on the application of SOM with a huge number of neurons and then grouping them using the HAC. The input layers correspond to the zonal and meridional velocities as well as the OW parameter. Then, at different sub regions statistics for the different parameters are evaluated including seasonal variations of clusters. If the objective of the Ms is to understand the mesoscale dynamics of the EMS the approach would be first to perform a temporal SOM analysis to the (ug,vg) velocities (or alternatively the MKE) to obtain the zones of co-variability. This would also provide the time series of the velocities in each of the patterns. This has to be done in conjunction with a spatial SOM that will give the main mesoscale structures in the basin. The BMUs of these spatial patterns decomposition will give the seasonality that the authors want to explain. However if the objective is to analyse the eddy activity in the area I suggest to change for the input data the EKE, MEKE and the OW parameter. In the paper no mention is given to which SOM they are applying nor the 5 clusters that they finally ended.

Minor concerns.

I assume that the data corresponds to daily velocities, but this is never stated in the Manuscript. Why using daily data and not weekly or monthly if the objective is to analyze mesoscale structures?

We thank the reviewer for this question. We could use weekly data because mesoscale structures extend from a few days to several weeks. However, since the current field in the Levantine Sea is characterized by a high Spatio-temporal variability (*Menna et al. 2012*), and eddies can appear/disappear or evolve quickly, we preferred to use daily data to avoid missing such short-time scale events.

Figure 1 and 2 can be merged.

Done

Figure 3. What are the units in the colorbar?

We added the values in the colorbars of the topological maps of each parameter (figure 2A,B,C)

Figure 4A. What is the message in this figure?

This figure shows the positions of the clusters in respect to the topological map

Figure 5. I suggest defining the areas directly with the SOM (see main concern)

We thank the reviewer for his suggestion. However, we find that this suggestion consists of an independent study, and that our choice of subregions of interest is justified in the paper.

To give a clear answer and to show how different the output is with such data re-organization, we performed the suggested classification, and we state the followed procedure below:

First, we structured the data set following each pixel's daily time series (from 1993 to 2018). In other terms, the data set corresponds to pixels of an image as rows and daily values as columns. We performed two independent classifications using 1) U, V and OW, 2) EKE, MKE, and OW. All variables were normalized to homogenize their weights and therefore their contribution to delimit clusters.

[Figure]

Using the new data set, one at a time, we trained a Self-Organizing map, and we proceeded in the same way with a HAC to cluster similar neurons in terms of daily time series. In both cases, 5 clusters were the best choice of cut-off level, at which the dissimilarity between clusters is important.

We reconstructed the temporal classification, and this showed in both cases non-homogeneous regions. In other terms, the clustering presents spatially intermittent regions that do not allow the definition of contingent areas.

This can be explained by the fact that these parameters do not reveal any clear spatio-temporal variability and does not allow to regroup clear regions with same variability.

This finding has been already highlighted by our approach, as seen in the paper (Figure 6 hal2), with the lack of any temporal periodicity or variability while looking at the succession of clusters inside each box.

In our paper, the choice of regions of interest was defined using the same approach as in (Barboni et al., 2021 based on the MDT. In addition to that, we considered in our paper the standard deviation to detect possible features that could be highly variable and unstable that does not appear in the mean signal of ADT (which was the case in CE, Shikmona, and Beirut boxes).

The boxes were spatially extended to include most of the variability within each feature as seen in the figure 3B. The spatial classification that we performed in this paper allowed to isolate different situations based on the parameters used and monitor them through time. The result in Figure 6 highlights the Spatio-temporal differences that occur within each box in response to different mesoscale activity.

Consequently, we hope that these new results are enough to convince the reviewer.

Figures 6-10. I don't understand the message behind these graphics.

By these figures we wanted to show the frequency variation of the five clusters in all the boxes. By figure 4 we showed that the cluster varies from one box to another and from one year to another. In figure 5 we showed the cluster that was daily most frequently cluster in every box and in figure 6 we showed that the energetic clusters (C1 C5) are increasing with time at the expense of weak current cluster C3. We moved the figures of the daily Mean Kinetic Energy and the trends of C1 and C4 to appendix, to reduce the number of figure.

Ln 90. Why using OW and velocities as input? %%

Using velocities was previously done in Jouini et al., 2016 to efficiently characterize the Sicily channel. However, since the Levantine is known for important eddy activity, we used additionally the OW parameter, which represents and allows to differ between stain and vortex-dominated areas. We wanted to base the study on the altimetric data that is with no gaps.

From U and V, we can estimate different mesoscale parameters such as MKE, using non-linear relationships that are conserved through the topology of the SOM.

Page 6. Why 1400 neurons?

We conducted several sensitivity tests to determine the size of the map. while increasing the size of the SOM map, we calculated the mean quantization error, which stands for the error between an observation and its Best Matching Neuron (closest neuron to this observation). For an increasing size of the SOM, this error keeps on decreasing. We chose 1400 neurons as a comprise between the quality of the SOM, its interpretability, and computational requirements.

[Figure]

Lines around 120. What do you mean with "SOM is well organized"?

We corrected it in the paragraph by adding that the topological map represents a clear gradient of U alongside clusters of intense V and OW. Since V and OW do not have the same gradient as U, it shows that these parameters follow a non-linear relationship.

Line 132, C$ instead of C5

Done

Line 156 "iso-MDT" . There are no isolines in this plot

We removed the "iso-MDT" lines term.

Section 3.1. See Main concerns.

---

## Author Response (AR2)

Dear Reviewer,

We appreciate your precious time in reviewing our work. We have carefully considered your comments below in black and tried to address them point-by-point in red. We refer to a modified pdf where all the modifications are highlighted in red.

1. I suggested to add the political boundaries on Figure 1. This was done, but the countries are not identified. I would suggest to identify at least those mentioned in the text.

   We agree with the reviewer, and therefore we added in figure 1 the countries that were mentioned in the paper.

2. L.49: I think that "currently" is the good word instead of "actually".

   Done.